# Testing Analytical Methods for Antibiotic Detection in *Tenebrio molitor* Larvae: A Controlled Feeding Trial

**DOI:** 10.3390/antibiotics14090909

**Published:** 2025-09-09

**Authors:** Tomke Asendorf, Christine Wind, Andreas Vilcinskas

**Affiliations:** 1State Institute for Chemical Analysis and Veterinary Diagnostics Freiburg, Am Moosweiher 2, 79108 Freiburg, Germany; 2State Institute for Chemical Analysis and Veterinary Diagnostics Karlsruhe, Weißenburger Str. 3, 76187 Karlsruhe, Germany; 3Branch for Bioresources, Fraunhofer Institute for Molecular Biology and Applied Ecology IME, Ohlebergsweg 12, 35392 Giessen, Germany; andreas.vilcinskas@ime.fraunhofer.de; 4Institute for Insect Biotechnology, Justus Liebig University Giessen, Heinrich-Buff-Ring 26, 35392 Giessen, Germany

**Keywords:** edible insects, industrial insect production, antibiotic residues, LC-MS/MS analysis, food safety monitoring

## Abstract

**Background:** As edible insects gain importance as sustainable protein sources, their integration into the food system requires that they meet the same safety standards as conventional animal products. This includes systematic testing for residues of pharmacologically active substances, such as antibiotics. To enable such monitoring, validated analytical methods for insect matrices are essential—but currently lacking. This study evaluates whether LC-MS/MS methods already validated for conventional animal products are suitable for detecting antibiotics in edible insects. **Methods:**
*Tenebrio molitor* larvae were fed wheat flour containing 10 mg of tiamulin or chloramphenicol and 31.3 mg erythromycin per 100 g flour. The antibiotics were mixed into the feed, and their homogeneity and stability were confirmed. After seven days of feeding and a 24-h fasting period, larval samples were analyzed by LC-MS/MS. **Results:** All three antibiotics were detected in the insects. After seven days, mealworms contained 6.8 ± 0.3 mg/kg tiamulin, 1.4 ± 0.2 mg/kg chloramphenicol, and 224.5 ± 111 mg/kg erythromycin. Following the 24-hour fasting period, concentrations declined markedly to 0.6 ± 0.03 mg/kg, 0.2 ± 0.002 mg/kg, and 130.5 ± 0.7 mg/kg, respectively. **Conclusions:** The detection of all three antibiotics demonstrates that existing LC-MS/MS methods can be applied to insect matrices. Owing to the small number of replicates and the exploratory nature of the trial, these residue levels should be interpreted qualitatively as a proof of concept. The study provides a reproducible model for further feeding trials and underscores the need for more comprehensive validation. Potential drivers of antibiotic misuse in insect farming are discussed as a basis for developing and expanding testing methods to ensure the food safety of edible insects.

## 1. Introduction

With the globally increasing demand for sustainable protein sources, edible insects are gaining growing attention from the scientific community, industry, and policymakers [1,2,3]. Insects for food and feed production offer numerous ecological advantages, such as high feed conversion efficiency, low greenhouse gas emissions, and the ability to be reared on organic waste streams [4,5,6]. As a potential alternative to conventional meat products, they are considered a promising solution for easing pressure on the environment and agricultural systems in the context of climate change and a growing global population [7,8,9].

At the same time, edible insects—as a novel food—face particular challenges regarding food safety and consumer acceptance [10]. Both aspects are essential and mutually reinforcing. Concerns about the uncontrolled use of substances such as antibiotics have already been raised in public discourse [11].

Some antibiotics, such as chloramphenicol, have indeed been used in insect rearing for research purposes—for example, in pest control or silk production [12,13]. Antibiotic residues may also enter insect production systems through contaminated substrates, such as manure [14,15]. Conversely, the use of insect frass as fertilizer could pose a risk for the development of antibiotic resistance in soil microbiomes if antibiotics were to be used in insect farming without proper regulation [16,17,18]. Improper feeding practices, such as the use of by-products from the meat industry, could likewise lead to contamination with antibiotics or other pharmacologically active substances [19].

Systematic screening of edible insects purchased from online retailers already revealed the presence of tiamulin in one sample [20]. However, it remains unknown whether pharmacologically active substances are intentionally used in insect farming, which compounds are relevant across commonly reared species, and whether current analytical methods are sufficient to ensure reliable detection. Because edible insects can harbor antibiotic-resistant bacteria [21,22,23,24,25], systematic monitoring of insect products for veterinary drug residues is essential. The European Food Safety Authority recommended such monitoring in 2015 [15], and EU regulations now require insects to be included in official control plans. Due to the currently low production volumes in Germany, the number of required samples remains limited: only two samples had to be analyzed in total across 2023 and 2024. However, this number is expected to increase in the coming years. By that time, at the latest, validated and appropriate analytical methods for the reliable testing of insects must be available [26].

To meet this need, the present study conducted a controlled feeding experiment with *Tenebrio molitor* larvae, administering tiamulin, chloramphenicol, and erythromycin. Tiamulin was included because it had previously been detected in an insect product [20]. Chloramphenicol has a history of use in insect rearing for scientific purposes and could therefore be relevant for edible insect production [12,27]. Erythromycin was selected due to the detection of resistance genes in marketed insects [28,29].

Our findings demonstrate the applicability of LC-MS/MS methods validated for conventional animal products to insect matrices. This proof-of-concept is an important step towards establishing validated methods for routine monitoring, which will be essential to guarantee the food safety of edible insects and to inform future regulatory frameworks.

## 2. Results

### 2.1. Impact of Antibiotics on Mealworm Weight

None of the tested antibiotics had a statistically significant effect on mealworm weight (Figure 1). Mean weights were slightly lower in the tiamulin group (0.74 ± 0.24 g, *n* = 20) compared to controls (0.82 ± 0.39 g, *n* = 20), but this difference was not significant (Mann–Whitney U = 135.0, *p* = 0.081). No significant weight differences were observed for chloramphenicol (0.74 ± 0.24 g vs. 0.83 ± 0.38 g; Welch’s t = 0.28, df = 33.92, *p* = 0.781) or erythromycin (0.72 ± 0.25 g vs. 0.83 ± 0.38 g; Welch’s t = −1.315, df = 36.82, *p* = 0.197).

GC–MS analysis confirmed that methanol used as a solvent for antibiotic preparation was not detectable in the feed. Control groups with and without methanol likewise exhibited no significant weight differences after the seven-day feeding period (0.82 ± 0.44 g with methanol vs. 0.85 ± 0.44 g without methanol; *p* = 0.544, Mann–Whitney U test; Figure 2; Appendix A).

### 2.2. Evaluation of Homogeneity and Stability of Antibiotic-Containing Feed Samples Using LC-MS/MS Analysis

Following Herrman et al., the homogeneity of the antibiotics in the flour was accessed by calculating the coefficient of variation (CV) of the samples (Table 1) [30].

The highest CV, and thus the lowest homogeneity, was observed in the flour spiked with chloramphenicol, with a coefficient of variation of 43.7%. The measured concentration of chloramphenicol in the flour was 119 ± 52 mg/kg. The coefficient of variation for the flour spiked with erythromycin was 19.5%, with a measured concentration of 185.8 ± 36.2 mg/kg. In the case of tiamulin and the 1:1000 diluted erythromycin, a coefficient of variation below 15% was achieved. For tiamulin, the CV was 13.3%, with a measured concentration of 325.9 ± 43.4 mg/kg in the flour. The coefficient of variation for the 1:1000 diluted erythromycin was 11.5%, with a measured concentration of 0.3 ± 0.03 mg/kg.

Furthermore, the antibiotics remained stable throughout the experiment under the given experimental conditions (Figure 3). In the control feed group, none of the administered antibiotics were detected. The raw LC–MS/MS data are provided in Appendix A.

### 2.3. Detection and Quantification of Antibiotics in Mealworms Using LC-MS/MS Analysis

All three antibiotics were detected and quantified in the mealworms after seven days of feeding and an additional 24 h withdrawal period, with the exception of the 1:1000 diluted erythromycin. In this group, only traces of erythromycin below the limit of quantification were detected immediately after the seven day feeding period. After 24 h of feed withdrawal, it was no longer detectable. All control groups remained blank.

Figure 3 presents the targeted and actual measured concentration of tiamulin, chloramphenicol, and erythromycin (undiluted) in the feed, as well as the final concentration detected in the mealworms. In the case of tiamulin the concentration detected in mealworms was 6.8 ± 0.3 mg/kg immediately after the seven-day feeding period and decreased to 0.6 ± 0.03 mg/kg after 24 h of feed withdrawal. This corresponds to a reduction of approximately 90% after 24 h. As previously described, chloramphenicol exhibited the highest inhomogeneity in the feed, which is evident in Figure 3 from the error indicators at Day 0. In mealworms, 1.4 ± 0.2 mg/kg of chloramphenicol was detected after the seven-day feeding period. After 24 h of feed withdrawal, the concentration decreased to 0.2 ± 0.002 mg/kg. This means that approximately 85% of the initially detected chloramphenicol was no longer quantifiable after the fasting period. The measured concentration of erythromycin in mealworms was 224.5 ± 111 mg/kg after seven days of feeding and decreased to 130.5 ± 0.7 mg/kg after 24 h of feed withdrawal (Figure 3). Due to an outlier in the measurement on day seven, the results are less clear for erythromycin. The reduction after 24 h of fasting ranged between 10% and 57%. However, overall, the detectable amount of erythromycin in mealworms also decreased. The raw LC–MS/MS data are provided in Appendix A.

## 3. Discussion

This study provides the first proof-of-concept that LC-MS/MS methods already established for other matrices can be successfully applied to edible insects for the detection of pharmacological compounds such as antibiotics. The feeding trial was designed to simulate realistic scenarios, guided by antibiotics previously detected in insects [20], documented uses in other species such as *Bombyx mori* [12], and associations with resistance genes in insect products [31]. The selection was also based on the methods available in the laboratory.

The design can serve as a template for future trials, but several refinements are advisable, such as homogenizing spiked feed using a laboratory mill. Incorporating blinding procedures during sample handling and outcome assessment would further strengthen experimental rigor. Moreover, the use of antibiotics in their commonly applied pharmaceutical formulations, as well as the inclusion of additional insect species and processing stages, should be considered to improve representativeness. As an initial exploratory study, the present work was necessarily limited by a small sample size, a restricted range of methods, and the focus on a single matrix; acknowledging these limitations will help guide the design of more comprehensive future investigations.

To identify which additional substances should reasonably be considered for further trials and ultimately included in monitoring, it is necessary to examine which pharmacologically active compounds may be relevant for edible insects. Insight can be gained by exploring the underlying motivations for the application of such substances in insects. Despite the absence of approved veterinary drugs for insects, several drivers for potential antibiotic use can be envisaged. These include growth promotion, prophylaxis against recurrent infections, and reduction of microbial contamination in the final product. Evidence from our study indicates that all three antibiotics tended to reduce larval weight rather than promote growth. Comparable negative outcomes have been reported in other insect species, including reduced reproduction, suppressed immunity, or altered microbial balance after exposure to pharmacologically active compounds [27,32,33]. Reduced feed palatability may also play a role, as suggested by previous studies on fumagillin and other compounds [32,34,35]. Nevertheless, reports such as Hirose et al. on positive effects of streptomycin on *Nezara viridula* show that performance-enhancing effects cannot be entirely excluded [36]. Another motivation for the use of pharmacologically active substances in insect farming may be their prophylactic application to prevent recurrent infections. Relevant insights into which infections are of particular concern in insect rearing are provided by studies such as those by Eilenberg et al. and Maciel-Vergara et al. [37,38]. Beyond prophylaxis against insect diseases during rearing, reducing the microbial contamination of the final product could also be a driving factor for the use of antibiotics. An initial risk assessment identifying human–pathogenic microorganisms potentially relevant in edible insect products was published by the EFSA in 2015 [15]. Additionally, Garofalo et al. provided an extensive literature review on the microbiota of insects covering the period from 2000 to 2019 [39]. As part of the present study, the generated samples were also subjected to microbiological analysis; however, no significant reduction in pathogenic microorganisms was observed in the antibiotic-treated groups compared to the control group (data not published). For this purpose, alternative decontamination methods—such as UV irradiation—may be more effective. The most recent approval of a mealworm product already includes this treatment, even though it is not directly used for microbial reduction in this case (Commission implementing regulation (EU) 2025/89).

A marked decrease in antibiotic concentrations was observed after the mandatory 24 h withdrawal period. As the compounds remained chemically stable in feed, this reduction must result from insect-internal processes such as excretion or metabolism. Systematic studies of pharmacokinetics in edible insects are lacking but urgently needed to determine whether residues persist in metabolized form. Analyses of excreta and tissue distribution would be essential to identify potential accumulation sites such as the fat body [40,41].

The current lack of approved veterinary drugs and defined maximum residue limits (MRLs) for insects leaves a regulatory gap. At present, insect products fall under the cascade principle (Regulation (EU) 2019/6), with default application of the lowest MRLs from other species (Implementing Regulation (EU) 2018/470). The prohibition of chloramphenicol and strict MRLs for tiamulin and erythromycin in livestock illustrate the relevance of this issue (Regulation (EU) No 37/2010). Since insects are consumed whole, alignment with tissue-specific MRLs from conventional animals may not be appropriate. Moreover, the potential role of antibiotics in selecting resistant bacteria in insects, as documented in several studies, underscores the urgency of establishing clear regulatory frameworks [29,31,42,43,44].

## 4. Materials and Methods

### 4.1. Antibiotics

For this study, pure antibiotic standards were used instead of complete pharmaceutical formulations, as the standards are more readily available and free from unpredictable additives. The antibiotics used were Tiamulin (Sigma-Aldrich, St. Louis, MO, USA), chloramphenicol (Sigma-Aldrich, USA), and erythromycin (Sigma-Aldrich, USA).

### 4.2. Supplementing Feed with Antibiotics

The tiamulin dose was 10 mg per 100 g of whole wheat flour (dmBio, Karlsruhe, Germany). In the absence of indications from the literature, the dose recommendation of the European Commission from Annex I of Article 35 of Directive 2001/82/EC (EMEA/V/A/137) was used as a guide. Based on Cappellozza et al., 10 mg chloramphenicol per 100 g flour was used [12]. For erythromycin, a concentration of 31.3 mg erythromycin per 100 g flour was employed. This was derived from the publication by Alippi et al. and adjusted to the weight of the mealworms determined in preliminary tests [45].

All antibiotics were initially dissolved in methanol (Merck KGaA, Darmstadt, Germany) at a concentration of 1 mg/mL. The dissolved antibiotics were then combined with whole wheat flour in a 2:1 ratio as follows: 20 mL of the tiamulin or chloramphenicol solution (1 mg/mL) was mixed with 10 g of flour. Following the evaporation of the methanol, the mixtures were homogenized with a mortar and combined with 190 g flour, yielding 200 g spiked flour each. For erythromycin, 62.5 mL solution (1 mg/mL) was mixed with 31.3 g flour, followed by evaporation and the addition of 168.7 g flour to a total of 200 g. To verify that no residual methanol remained in the flour after evaporation, control group samples (with and without methanol treatment) were analyzed using headspace GC-MS (Thermo Trace 1310 GC, ThermoFisher, Waltham, MA, USA; detection limit: 50 mg/kg).

### 4.3. Feeding Experiment

Mealworms (larvae of the mealworm beetle *Tenebrio molitor*) were obtained from diemehlwurmfarm (Gronau, Germany) and were reared at the State Institute for Chemical Analysis and Veterinary Diagnostics Freiburg. A total of ten boxes, each measuring 15 × 11.5 × 7 cm, were used. Each box was filled with 10 g of mealworm and 10 g of whole wheat flour. For each of the three antibiotics (tiamulin, chloramphenicol, and erythromycin), two boxes were filled with spiked flour (six boxes in total). Four boxes served as control groups. In two of them, the mealworms were fed only with flour, while in the other two, the flour was supplemented with methanol, which was then allowed to evaporate, mirroring the antibiotic treatment conditions. To assess the potential impact of antibiotics on weight development, mealworms were weighed before and after seven days of treatment. For each group, 10 batches of 10 randomly selected larvae were weighed to obtain a representative average. All larvae were incubated at 27 °C and 60–70% humidity in an incubator (Binder, Tuttlingen, Germany) for seven days. After this period, one box of each substrate type was frozen at −20 °C. The remaining boxes were sieved to extract the larvae from the substrate, and the larvae were fasted in the empty boxes for another 24 h, and then frozen at −20 °C. All larvae were kept at −20 °C for at least 24 h before processing to ensure all larvae were killed. The four boxes frozen after seven days were sieved to separate larvae from the substrate, as with all other boxes. The larvae were rinsed under running tap water, dried on paper towels, and homogenized in an IKA Tube Mill 100 control (IKA, Staufen im Breisgau, Germany). Homogenates prepared from the tiamulin, chloramphenicol, erythromycin, and control groups after 0 and 24 h fasting were stored at −20 °C.

### 4.4. Evaluation of Homogeneity and Stability of Antibiotic-Containing Feed Samples Using LC-MS/MS Analysis

Homogeneity of the antibiotic–feed mixtures was assessed by analyzing five samples of each mixture immediately after preparation. Stability was evaluated by analyzing residual feed samples at the end of the feeding trial. Specifically, sieved feed residues from each experimental group (tiamulin, chloramphenicol, and erythromycin) were examined for both the 0 h and 24 h fasting groups. The feed analysis was conducted following the method 14.1.5 of the Association of German Agricultural Analytic and Research Institutes (Verband deutscher landwirtschaftlicher Untersuchungs und Forschungsanstalten e.V. (VDLUFA e. V.)) [46]. The analyses were performed at the Department of Pharmaceutical Residues of the State Institute for Chemical Analysis and Veterinary Diagnostics, Karlsruhe, Germany. The only deviation from the standardized method was the use of a drying temperature of 40 °C during sample concentration.

Samples were extracted using McIlvaine buffer with Na-EDTA (pH 4) and purified via SPE using Oasis HLB 6cc 200 mg cartridges (Waters, Milford, MA, USA). The analysis was conducted using an Agilent 6460 Triple Quadrupole LC/MS system (Agilent, Santa Clara, CA, USA). Data evaluation was performed using the Agilent MassHunter 10 software (Agilent, USA).

The concentrations of all three antibiotics used exceeded the upper calibration range of the analytical method. Accordingly, the sample weight of the feed samples was adjusted, and the samples were diluted as follows: For the flour samples spiked with tiamulin, 0.5 g instead of 5 g was weighed per sample. Before solid phase extraction (SPE), a 1:20 dilution was performed. For the flour samples spiked with chloramphenicol, 0.1 g instead of 10 g was weighed per sample. Residues were collected using 2 mL instead of 0.5 mL extraction solvent (water/acetonitrile, 80/20, *v*/*v*), and the extract was diluted 1:10 prior to SPE measurement. For flour samples spiked with erythromycin, 0.5 g instead of 5 g was weighed, and the extract was diluted 1:50 before SPE (0.2 mL extract + 9.8 mL McIlvaine buffer with Na-EDTA, pH 4). Due to the high concentrations and to establish an initial estimate of the detection limit, the amount of erythromycin in the feed was reduced by a factor of 1000 in a second feeding trial, resulting in 31.3 µg erythromycin per 100 g flour. In this case, 1 g of the erythromycin–flour mixture was weighed for LC-MS/MS analysis.

### 4.5. Detection and Quantification of Antibiotics in Mealworms Using LC-MS/MS Analysis

The analyses were performed at the Department of Pharmaceutical Residues at the State Institute for Chemical Analysis and Veterinary Diagnostics, Karlsruhe, Germany. The method used is validated according to the Commission Implementing Regulation (EU) 2021/808 for various foodstuffs of animal origin and is used in routine testing for antibiotic residues. The adaptation of this method for insects is not part of the present study. Therefore, only a brief summary is provided here (for detailed information, see Appendix A). Homogenized and pre-weighed samples were thawed. After adding 10 mL McIlvaine buffer with Na-EDTA (pH 4), the samples were vortexed for approximately 1 min, shaken for 10 min, sonicated for 5 min, and centrifuged at 3148 g for 10 min. The supernatant was filtered through a folded filter and collected in a 50 mL centrifuge tube. The extraction was repeated with 5 mL McIlvaine buffer with Na-EDTA (pH 4), and the filter was rinsed with an additional 2 mL McIlvaine buffer with Na-EDTA (pH 4). The entire extract was then subjected to SPE. An OASIS HLB SPE cartridge (6 mL, 200 mg) was conditioned with 6 mL methanol and 6 mL deionized water. The extraction solution was loaded onto the cartridge, washed with 6 mL ultrapure water/methanol (95/5, *v*/*v*), and dried under vacuum for 10 min. Elution was performed with 6 mL of methanol into a centrifuge tube. The eluate was evaporated to dryness under a nitrogen stream at 40 °C. The dry residue was reconstituted in 1 mL ultrapure water/acetonitrile (90/10, *v*/*v*) using vortex mixing and ultrasonication. After centrifugation at approximately 15,000 rpm, the solution was transferred to an HPLC vial for LC-MS analysis or stored frozen until measurement. The identification of target analytes in the samples was based on mass spectrometric data, specifically the detection of at least two MS^2^ fragment ions, and retention time comparison with corresponding standards. Each sample was initially processed and analyzed once. If a target analyte was detected, its concentration was determined using a matrix calibration curve established with control samples. In cases of critical initial findings, multiple additional sample preparations were performed and analyzed. To generate the matrix calibration curve, blank samples (e.g., five replicates) were fortified with different concentrations of the analyte, processed, and analyzed. In the case of chloramphenicol and undiluted erythromycin, the measured concentrations exceeded the calibration curve. Therefore, a 1:100 dilution of the final extracts (10 µL extract + 990 µL water/acetonitrile) was performed for quantification. Since this dilution was still insufficient for erythromycin quantification, the already 1:100 diluted extract was further diluted 1:5. The final result was obtained by averaging the individual measurements.

## 5. Conclusions

The results of this study demonstrate that analytical methods already established for conventional animal-derived foods can also be applied to detect antibiotics such as tiamulin, chloramphenicol, and erythromycin in *Tenebrio molitor* larvae. The experimental design and outlined considerations offer a first step towards analyzing antibiotic residues in edible insects. As an initial exploratory study, it was necessarily limited by a small sample size, a restricted range of methods, and the focus on a single matrix; nevertheless, the design and results provide a useful basis to guide future investigations. Overall, the findings contribute to building a scientific foundation to ensure that edible insects are, and remain, a safe food source.

## Figures and Tables

**Figure 1 antibiotics-14-00909-f001:**
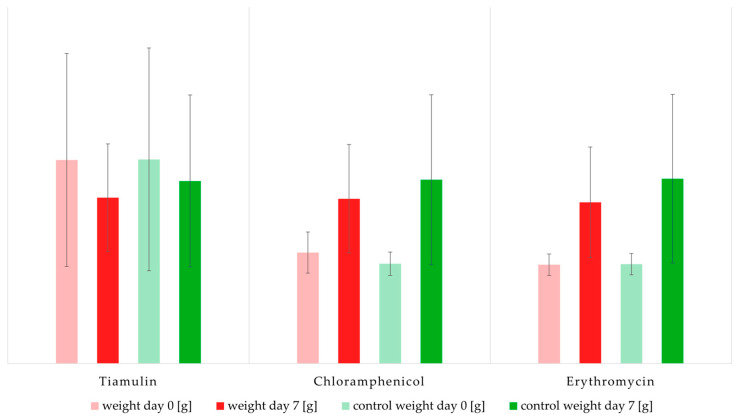
Weight development of mealworms after seven days of antibiotic treatment (red) compared with untreated controls (green). Data are presented as mean ± standard deviation from 10 groups of 10 larvae each. Statistical analysis revealed no significant differences between treatments and controls.

**Figure 2 antibiotics-14-00909-f002:**
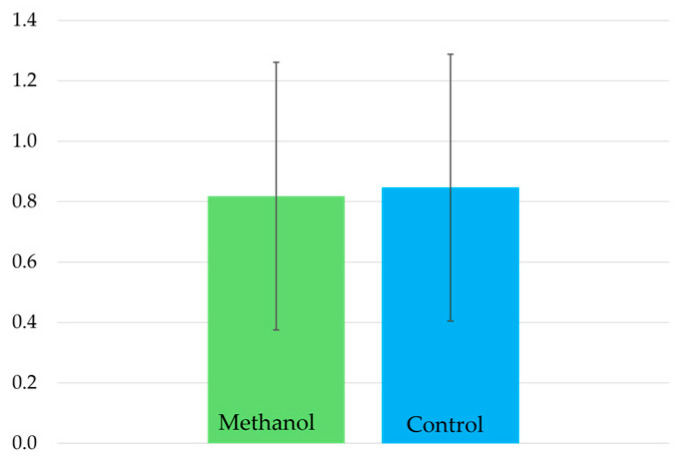
Weight (g) of mealworms after seven days in flour previously treated with methanol (green) compared to untreated flour (blue). Values represent mean ± standard deviation.

**Figure 3 antibiotics-14-00909-f003:**
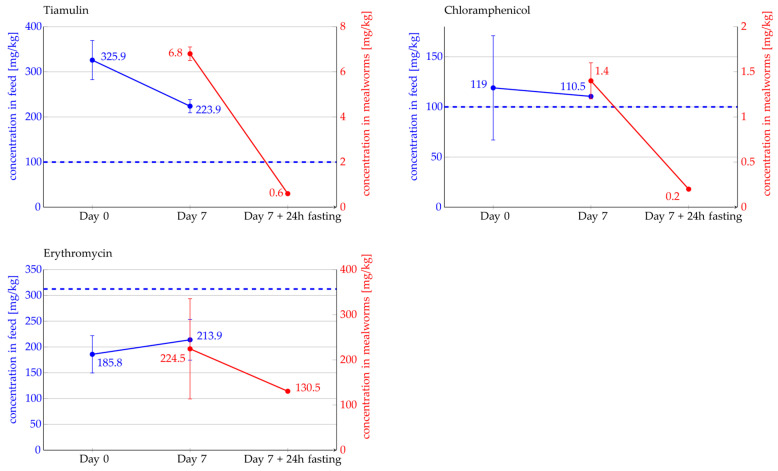
Concentration of tiamulin, chloramphenicol, and erythromycin (undiluted) measured in feed (left y-axis, blue) and in mealworms (right y-axis, red). Error bars indicate standard deviation (SD). The blue dashed horizontal line represents the target feed concentration (left y-axis).

**Table 1 antibiotics-14-00909-t001:** Homogeneity of antibiotics in feed.

	Tiamulin	Chloramphenicol	Erythromycin	Erythromycin/1000
Average, $\bar{X}$ [mg/kg]	325.9	119	185.8	0.3
SD [mg/kg]	43.4	52	36.2	0.03
coefficient of variation (CV) [%]	13.3	43.7	19.5	11.5

## Data Availability

The original contributions presented in this study are included in the article/Appendix A. Further inquiries can be directed to the corresponding author.

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
