# Peer review of "Testing Analytical Methods for Antibiotic Detection in Tenebrio molitor Larvae: A Controlled Feeding Trial"

_antibiotics, 2025, doi:10.3390/antibiotics14090909_

Round 1
Reviewer 1 Report
Comments and Suggestions for Authors
Dear authors,
Thanks for this interesting world. You will find some minor suggestion in my report

Reviewer 2 Report
Comments and Suggestions for Authors
The manuscript titled “Testing analytical methods for antibiotic detection in Tenebrio molitor larvae: A controlled feeding trial” investigates the persistence of antibiotics in live food organisms, specifically Tenebrio molitor larvae. The authors administered three commonly used antibiotics—tiamulin, chloramphenicol, and erythromycin—for one week and assessed their persistence using LC-MS/MS. They reported that all three antibiotics were detected in the larvae, although residue levels decreased significantly after fasting. The study provides a reproducible model for feeding trials and highlights factors contributing to antibiotic misuse in insect farming, offering a foundation for improved testing methods to ensure the safety of edible insects.
The manuscript is interesting and significant for antibiotic measurement in food systems. The findings are important, and the study design is sound. Although the English language is good, the description does not meet scientific standards. Therefore, the manuscript should be revised to strictly follow the standard scientific format, including a clear and logical structure (Abstract, Introduction, Materials and Methods, Results, Discussion, and Conclusion) and concise, precise language. I recommend a major revision and suggest that the complete manuscript be rewritten with the assistance of an experienced scientific writer.
Specific comments:
- Abstract: The abstract is general. The authors need to add specific values in methods and results, e.g., dose of antibiotics administered and the quantity detected in the sample.
- Keywords: Please avoid words used in the title.
- Introduction: Too many small paragraphs. Please summarize the information and arrange 4/5 paragraphs.
- Introduction: Please avoid frequent use of 'our' or 'we' in scientific writing, as it breaks the norms of formal description. Instead, rephrase sentences to use an indirect form. See line number: 83, 88, 90, 91, 105, 283, 284, 285, 286, 292, 333, 334, 336, 367, 368, ....
- The authors should improve the Introduction section by adding the research gap and the importance of the present study at its end.
- Results Line: 108; I recommend presenting the supplementary data as main data, as the amount of data is not too extensive.
- Results: 2.2: valuation of homogeneity and stability of antibiotic-containing feed samples using LC-109 MS/MS analysis: The authors described methods in this section (Lines: 111-126). This requires massive change.
- Results: 3. Detection and Quantification of antibiotics in mealworms using LC-MS/MS analysis, Lines: 143-151: Why did the authors described the methods here?
- Discussion: The Discussion section contains too many short paragraphs. The authors should rewrite it by summarizing the information into 5–6 well-structured paragraphs, focusing on the observed findings, the reasons behind them, comparisons with previous reports, and explanations for any agreements or discrepancies with those reports.
- Materials and Methods: The authors should rewrite this section to present the methods concisely and specifically. It currently contains introductory discussions, for example, in lines 290–315.
- Conclusions: The authors are advised to improve this section by including the key findings of the study and noting any limitations, if applicable.
Reviewer 3 Report
Comments and Suggestions for Authors
This manuscript highlights the importance of monitoring antibiotics in edible insects, which are considered a promising alternative protein source. Therefore, systematic testing for residues using validated analytical methods is essential. The study presents an LC–MS/MS-based analytical method and demonstrates its reproducibility for future feeding trials. While the method is feasible and employs advanced techniques, the manuscript requires major revision before publication.
-
The background and discussion sections contain redundant information and overly verbose descriptions. These parts should be revised to be more concise and focused.
-
In Section 2.1, the inclusion of a figure or table would make the presentation clearer and easier to follow.
-
For the results, providing LC–MS/MS data—such as HPLC profiles and MS spectra—would significantly strengthen the persuasiveness of the findings.
In summary, the manuscript is verbose and lacks clarity in its current form. Improving conciseness, adding supporting figures/tables, and presenting detailed LC–MS/MS data would greatly enhance its readability and impact. I recommend publication only after major revision.
Round 2
Reviewer 2 Report
Comments and Suggestions for Authors
The authors have improved the manuscript according the reviewer's comments and now it can be accepted.
Reviewer 3 Report
Comments and Suggestions for Authors
The revised manuscript is much improved compared to the initial version and addresses all of the concerns raised in my previous review. Therefore, I recommend it for publication in its current form.